# Transition cow health and management in pasture-based dairy herds: A farmers' survey

**Louise Horan** [1,2], **Joseph Patton**[1], **Conor G. McAloon**[2], **Ángel García-Muñoz** [3], **Áine Regan**[4], **John F. Mee**[1], **Ainhoa Valldecabres** [1,3]*

1 Animal and Grassland Research and Innovation Centre, Teagasc Moorepark, Fermoy, Co. Cork, Ireland, 2 School of Veterinary Medicine, University College Dublin, Belfield, Dublin, Ireland, 3 School of Veterinary Medicine, Universidad Cardenal Herrera-CEU, CEU Universities, Valencia, Spain, 4 Department of Agri-food Business & Spatial Analysis, Teagasc Mellows Campus, Athenry, Co. Galway, Ireland

* ainhoa.valldecabres@uchceu.es

## Abstract

Seasonal-calving pasture-based systems characterize Irish dairy production. During the dry period, cows are housed and offered predominantly grass silage, providing unique transition cow management opportunities. This study aimed to describe transition period disease incidence and management strategies reported by farmers, and to evaluate their associations with herd size and calving pattern to inform and guide research activities and national advisory. An online survey distributed amongst 3,899 Teagasc Technical Dairy Advisory clients yielded 525 responses suitable for analysis. Results are presented for all respondents, by herd size and by the two most common calving systems (spring- [84.3%; 439/521] and split-calving [12.9%; 67/521]). Disease incidence was reportedly highest in cows within their first 3 weeks postpartum (58%; 299/519), in cows calving at the end of the calving season (48%; 245/510) and in multiparous cows (52%; 266/513). Respondents reported treating >3% of their herd for milk fever (23%; 120/521) and retained placenta (13%; 68/518), and <1% of their herd for grass tetany (82.6%; 419/507) and ketosis (72.7%; 368/506). Regarding management, dry cow body condition monitoring (73%; 365/497), dry cow mineral supplementation (61%; 304/497), and Ca supplementation at calving (61%; 314/487) were most commonly reported. Other milk fever prevention strategies supported by research in other production systems were not commonly reported (low K [20%; 101/497] and negative dietary cation-anion difference diet [6%; 31/497]). The odds of reporting keeping records of antibiotic treatment for milk fever were higher (OR = 3.20) for farmers from small compared to large herds. In conclusion, responses to our survey suggest that milk fever is a transition cow health concern in Irish dairy farms. Efforts should be devoted to enhance farmers' uptake of existing research-supported prophylactic strategies for milk fever and to optimize commonly reported management strategies in the Irish dairy production context.

## Introduction

The transition period, encompassing the few weeks before and after calving in dairy cows has been a focus of research over the last few decades. This is not surprising given the range of

**Data Availability Statement:** All relevant data are within the manuscript and its Supporting Information files. Survey response details are confidential as our survey was not anonymous

(Lines 128-129), and permission was only sought form the respondents for the purpose of this study and contact for other specific study.

**Funding:** Funding for this project was provided by Teagasc (RMIS 1765) and awarded to AV, project coordinator and principal investigator of the above project. Please replace "(RMIS 1765)" with "(Project Number 1765).

**Competing interests:** The authors have declared that no competing interests exist.

physical (physiological, immunological and metabolic) and environmental changes which challenge cows' homeostasis and homeorhesis, often turning into disease and ultimately impairing cows' welfare and production performance [1]. Despite the large amount of research conducted in the transition period and management strategies for its optimization, it remains a challenge to dairy production. The lack of a single definition for the transition period, as well as varying farmers' attitudes towards management and veterinarian involvement have been described as barriers to transition cow health and management improvement by a study involving Canadian farmers of confined herds and veterinarians [2]. Redfern et al. [3] interviewed farm advisors and reported that advisors were not providing farmers with focussed advise due to time constraints and fear of responsibility, among others. This lack of focussed advice being given to farmers may also restrain the improvement on transition cow health and management.

While the challenges faced by housed and grazing cows during the transition period may be similar, system-level differences determine the management possibilities and the occurrence of specific diseases for these two production systems. As discussed in a review by Roche [4], there is wide variability among pasture-based dairy production systems, potentially leading to problems unique to each system. In Ireland, dairy herds are predominantly intensive spring-calving herds in which cows graze the majority of their lactational feed requirements and are housed and fed conserved forages during the dry period in the winter months. Forage, mostly grazed pasture, makes up 95% of the Irish dairy cows' diet [5], creating a need for bespoke transition cow management. Nevertheless, limited transition cow health research has been conducted in this context and there is a lack of national-level disease incidence and management data which is needed to characterize and benchmark against current scientific recommendations for transition cow health and management strategies implemented in this production system.

Quantitative surveys have been used to describe transition cow disease incidence and management strategies in other dairy production systems [6, 7]. However, to the best of our knowledge, the only available survey associated with the Irish dairy cow transition period focuses on calving and colostrum management briefly describing pre-calving nutritional management in Irish dairy herds [8]. Therefore, the purpose of this study was to describe farmers' reported disease incidence and management strategies implemented during the transition period, and to quantify their associations with herd size and calving pattern to inform and guide research and advisory activities in transition cow health and management in Irish dairy farms.

## Materials and methods

The present study was approved by the University College Dublin Human Research Ethics Committee–Sciences (LS-LR-22-180; HREC-LS). A tick the box question at the beginning of the survey was used to obtain written consent from respondents to use data provided in the survey and data available in their Irish Cattle Breeding Federation (ICBF) profiles for the purpose of this study.

### Study population

Teagasc Technical Dairy Advisory clients were the target population of this observational study. Teagasc is the Agriculture and Food Development Authority in the Republic of Ireland and is composed of three main pillars: research, education and advisory/extension. Irish farmers voluntarily sign up to the advisory service which aims at disseminating independent, research-driven technical advice and support to clients. This is achieved by means of offering monthly farmer discussion groups, regular on-farm consultations, and provision of decision

support packages and printed/audio material. At the time of the study, a total of 3,899 nation-wide Irish dairy farmers were clients of the Teagasc Technical Dairy Advisory services and had provided consent for being contacted for research purposes; this represents 25.5% of Ireland's dairy farmers in 2022 [9]. The wider dairy farming community could not be targeted in this study due to limitations on personal data access for the researchers.

## Survey design and distribution

An online survey was designed to collect information on Irish dairy farmers' transition period perception, disease incidence and implemented management strategies. For the purpose of this study, focus is given to the disease incidence and implemented management strategies survey sections. Questions were modified according to Teagasc dairy advisors' suggestions, and the survey was pilot tested on five people who were either dairy farmers or dairy farm managers to assess its effectiveness and estimate the time to completion. The survey was administered using SurveyMonkey (SurveyMonkey Inc., Palo Alto, CA). At the beginning of the survey the transition period was defined as "late dry (late pregnancy if primiparous) to early lactation period" to provide context to respondents. The survey included 18 questions; questions were a mixture of closed (multiple choice; n = 14), open-ended (n = 3) and multiple choice with a comment field to allow respondents to provide a response that was not listed (n = 2). The first question asked respondents to confirm consent to data usage by ticking a box. The second question asked farmers for their herd number for the purpose of data extraction from the ICBF database and was followed by two questions relating to interest in participating in a subsequent on-farm study. Given their lack of association to the survey results, these two questions are not included in the survey available as supplementary material (S1 Table). Afterwards, three questions gathered farmers opinions and perception of the transition period, and the remaining questions (n = 11) gathered information to meet the objectives of this study regarding respondent demographics (n = 2), disease incidence (n = 6) and management strategies (n = 4; S1 Table).

The link to the online survey along with an explanatory message were distributed by text message to Teagasc technical dairy advisory clients (n = 3,899) on the 28th September 2022. A reminder text was sent on the 4th October 2022 and the survey was closed for responses 12 days after its opening.

## Data processing and analysis

Survey responses were exported to Excel (Excel 2013; Microsoft Corp.) for analysis. Survey responses are confidential. Four respondents answered the survey twice; the survey response with the highest level of completion or that provided in the first attempt, if both responses had the same level of completion, were used in the study. Seventy-two respondents skipped every survey question after providing consent for data usage for research purposes and were not included in the analysis. Responses were checked for signs of bot activity before data analysis by checking timestamps to ensure no respondents completed the survey abnormally fast and by checking responses for any illogical or repeated statements [10].

Answers in the open-ended comment fields of some of the multiple-choice questions were placed into new or already existing categories within the question for data analysis and summarization. Similarly, some answers to the same question were grouped; given the prevalent inclusion of Mg in pre-made mineral mixes used in Ireland (ie Reardon et al. [unpublished]), responses reporting the provision of dry or fresh cow minerals were combined with those reporting Mg supplementation to dry or fresh cows in respective categories named "Mg and/or other mineral supplementation". The two categories "high-risk cows Ca supplementation at

calving" and "all cows routine Ca supplementation at calving" were combined into "Ca supplementation at calving". Where respondents had the option of selecting an answer or not, the selection of the answer was coded as "yes" and the lack of selection was coded as "no"; consequently, answers such as "I don't keep records of this disease" and "no, I don't get advice from any of the above" were no longer considered in the analysis as these were already regarded in the above described code. Given the systematic provision of concentrates during milking to lactating cows [11], responses reporting the provision of feeds other than silage to fresh cows were not considered in this study. Answers to reported herd disease treatment incidence were summarized as "above" or "below" herd alarm levels previously described in a review by Lean and DeGaris [12]; where the described herd alarm level did not coincide with the answer options specified in the survey, the closest category was referred instead. Only diseases with at least 20% of reported treatments at each side of the herd alarm threshold were evaluated for their association with herd size and calving pattern.

Respondents were classified by herd size using information from the Teagasc advisory and ICBF databases, categories were defined based on the Irish national dairy herd average size (93 cows; [16]) as large (>150 cows), above average (100–150 cows), average (60–100 cows), or small (<60 cows); herd size information was obtained for 510 of the respondents. Respondents were also classified by calving pattern using the information provided in the survey (spring-calving only, autumn-calving only, split-calving, or all year-round calving) with only the two most commonly reported calving patterns being used in analysis (spring- and split-calving). Further herd-level descriptive information (305-day milk yield and calving interval) was obtained from the ICBF database.

Summary statistics were produced using the MEANS and FREQ procedures of SAS (Version 9.4; SAS Institute Inc., Cary, NC). Univariate logistic regression models were used to evaluate the association between reported disease treatment incidence and implemented management strategies with herd size or calving pattern using the GENMOD procedure of SAS. Statistical models included the logit link function and the Tukey-Kramer adjustment to account for multiple pairwise comparisons (herd size models). Reported odds ratio (OR) represent the ratio for the odds of "yes" vs. "no" answer to each question for respondents belonging to different herd size or calving pattern categories, taking as a reference the most prevalent categories (large herd size and spring-calving). Only OR at $P \leq 0.05$ for the comparison are reported in the manuscript. Considering each respondent did not answer every question of the survey, the number of respondents per question (and answer) is provided as appropriate.

## Results

A total of 601 survey responses were received between 28th September and 10th October (2022); yielding a survey response rate of 15.4%. Excluding the duplicated (n = 4) and blank responses (n = 72), 525 responses were available for analyses. Geographical distribution by county of survey respondents providing a valid Eircode is presented in Fig 1. On average, it took respondents 14 minutes to complete the survey. Responses are reported for all respondents (n = 525), by herd size (large: n = 154, above average: n = 134, average: n = 148, or small: n = 74) or calving pattern for the two most common calving systems (spring-calving: n = 439, or split-calving: n = 67). Denominator values are shown for each question and answer; lower denominator values indicate questions or answers skipped by some respondents.

### Study population

Overall, respondents median herd size was 110 cows (interquartile range [IQR] = 78–162 cows) and mean herd size was 135 cows. Respondents mainly had spring-calving herds

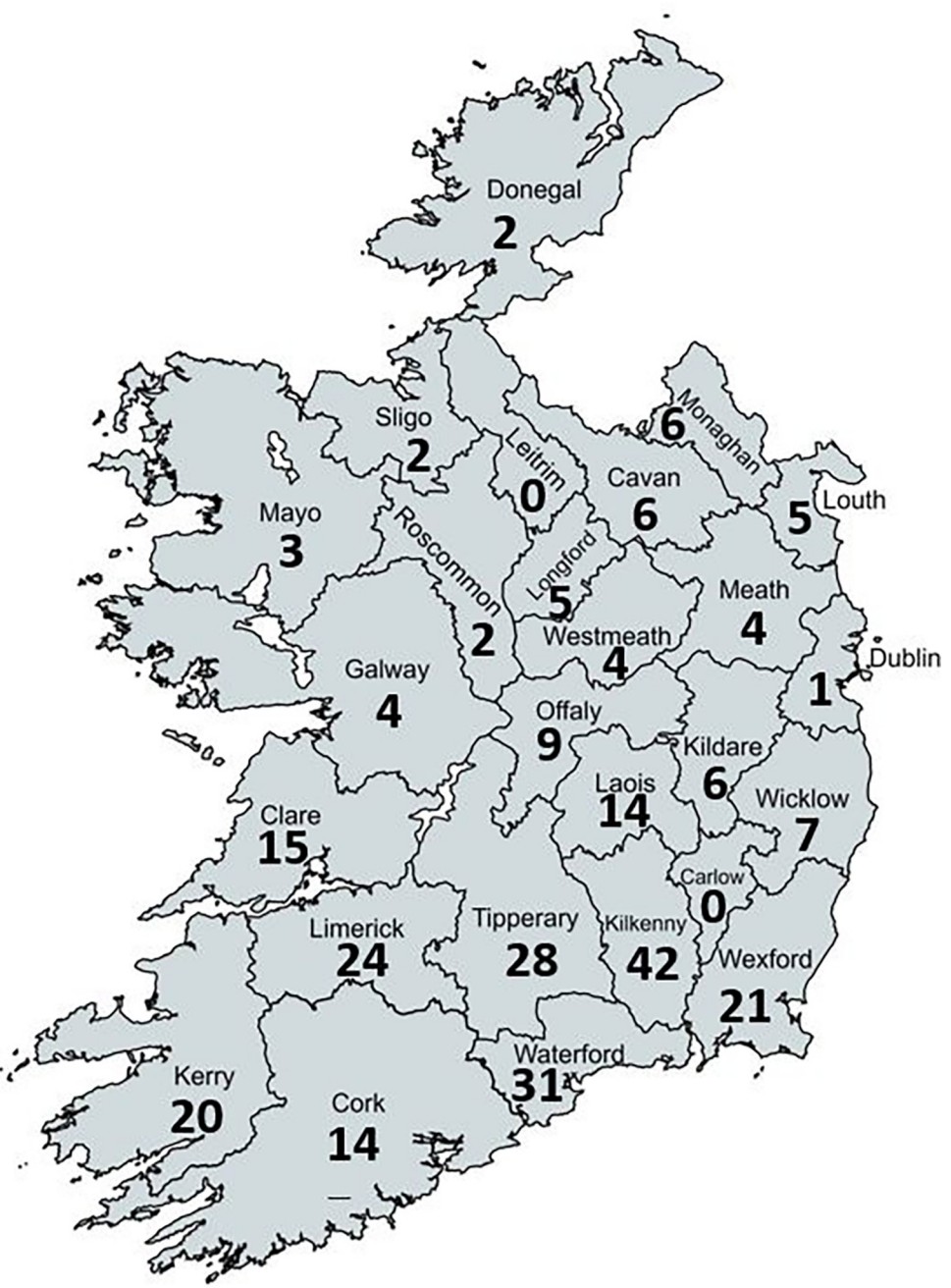

**Fig 1. Geographical distribution by county of survey respondents across the Republic of Ireland (n = 408 respondents with valid Eircodes).** Created with MapChart.net and adapted by authors.

(84.3%; 439/521) whilst the remainder operated split-calving (12.9%; 67/521), all year round calving (2.3%; 12/521), or autumn-calving (0.6%; 3/521) herds. For farmers with an active ICBF account with relevant data available, mean 305-day milk yield was 6,857 L (IQR = 6,111–7,162 L; n = 237) and mean calving interval was 377 days (IQR = 367–381 days; n = 323) for 2022. Based on the amount of bought-in feed per cow per year, farmers classified themselves as high-input (>1 tonne of bought-in feed/cow; 51.6% [268/519]), low-input (≤1 tonne of

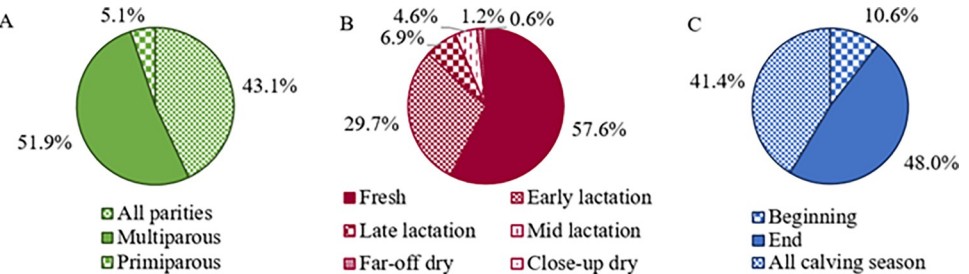

**Fig 2.** Reported distribution of highest disease incidence by cow parity (A; n = 513), stage of lactation (B; n = 520) and stage of calving season (C; n = 510) for all respondents to a transition period survey in Ireland.

bought-in feed/cow; 47.8% [248/519]), or zero-grazed grass fed all of the time (0.6%; 3/519). Herd descriptions by herd size and calving pattern are provided in S2 Table.

## Disease incidence

The complete distribution of reported disease incidence according to stage of calving season, stage of lactation and parity by herd size and calving pattern is presented in S3 Table. Incidence of disease was reported to be highest in freshly calved cows (first 3 weeks after calving; 57.6% [299/519]) and in multiparous cows (51.9%; 266/513). Respondents reported that disease incidence was highest among cows calving at the end of the block calving season (with late calvers; 48.0% [245/510]). However, a substantial cohort of respondents, indicated that problems arise during all of the calving season regardless of the stage (41.4%; 211/510) and that disease equally affects both, primiparous and multiparous cows (43.1% [221/513]; Fig 2). Table 1 shows the complete distribution of the reported proportion of cows treated by condition, herd size and calving pattern. Overall most farmers reported treating ≤3% of their herd for milk fever (77.0%; 401/521) and retained placenta (held cleaning; 86.9% [450/518]), and <1% of their herd for grass tetany (82.6%; 419/507), ketosis (72.7%; 368/506), displaced abomasum and/or digestive problems (71.5%; 373/522), and metritis (52.4%; 263/502) on an 'average' year on their farm. The odds of farmers from split-calving herds to report treating >3% of the herd for milk fever were 1.8 times those of farmers from spring-calving herds (OR [95% CI] = 1.78 [1.02–3.12]; $P$ = 0.042). The association between reported incidence of other diseases and herd size or calving pattern was not evaluated given the limited number of farmers reporting to treat a proportion of animals above the herd alarm levels described by Lean and DeGaris [12].

## Perceived disease importance

The complete distribution of perceived disease importance as reported by herd size and calving pattern is described in S4 Table. Based on incidence and impact in their herd, most of the respondents indicated that occasional cases without major effect on herd performance were observed for milk fever and/or downer cow (73.0%; 381/522), metritis (72.2%; 374/518), ketosis (70.0%; 319/523), retained placenta (held cleaning; 69.1% [357/517]), and displaced abomasum and/or digestive problems (61.9%; 88/522). However, a substantial proportion of the respondents indicated that milk fever was a significant (regularly treating severe cases with some cows lost/culled) or routine (regularly treating cows to control issues) problem in their herds (15.7%; 82/522). Subclinical hypocalcaemia was reported as a significant or routine problem in some herds (9.4%; 49/522), nevertheless, 20.7% (107/517) of farmers reported not knowing if subclinical hypocalcaemia was a problem in their herd.

**Table 1. Reported proportion of respondents' herds treated for health conditions on an "average" year (% of respondents to a transition period survey in Ireland).**

| Condition and treated cows | Herd size[a] | | | | Herd calving pattern[a] | | |
|---|---|---|---|---|---|---|---|
| | Large | Above average | Average | Small | Spring-calving | Split-calving | All |
| Milk fever | n = 154 | n = 133 | n = 146 | n = 74 | n = 437 | n = 67 | n = 521 |
| >10% | 2.0 | 1.5 | 2.7 | 1.4 | 1.8 | 3.0 | 1.9 |
| 7 to 10% | 4.6 | 5.3 | 2.1 | 1.4 | 3.7 | 4.5 | 3.8 |
| 4 to 6% | 14.3 | 18.8 | 15.1 | 24.3 | 16.0 | 25.4 | 17.3 |
| 1 to 3% | 55.2 | 48.9 | 48.0 | 33.8 | 49.0 | 43.3 | 48.4 |
| <1% | 24.0 | 25.6 | 32.2 | 39.2 | 29.5 | 23.9 | 28.6 |
| Retained placenta (held cleaning) | n = 154 | n = 133 | n = 146 | n = 71 | n = 435 | n = 67 | n = 518 |
| >10% | 0.0 | 1.5 | 0.0 | 0.0 | 0.5 | 0.0 | 0.4 |
| 7 to 10% | 2.6 | 2.3 | 2.1 | 0.0 | 2.3 | 1.5 | 2.1 |
| 4 to 6% | 11.0 | 10.5 | 13.7 | 5.6 | 10.1 | 13.4 | 10.6 |
| 1 to 3% | 60.4 | 54.1 | 50.0 | 56.3 | 53.8 | 64.2 | 55.6 |
| <1% | 26.0 | 31.6 | 34.3 | 38.0 | 33.3 | 20.9 | 31.3 |
| Metritis | n = 148 | n = 127 | n = 141 | n = 72 | n = 424 | n = 63 | n = 502 |
| >10% | 0.0 | 0.8 | 0.7 | 0.0 | 0.2 | 1.6 | 0.4 |
| 7 to 10% | 2.0 | 1.6 | 2.8 | 1.4 | 2.4 | 0.0 | 2.0 |
| 4 to 6% | 8.8 | 7.9 | 8.5 | 5.6 | 8.5 | 7.9 | 8.2 |
| 1 to 3% | 45.3 | 39.4 | 29.8 | 29.2 | 34.9 | 47.6 | 37.1 |
| <1% | 43.9 | 50.4 | 58.2 | 63.9 | 54.0 | 42.9 | 52.4 |
| Displaced abomasum | n = 153 | n = 134 | n = 147 | n = 74 | n = 438 | n = 66 | n = 522 |
| >10% | 0.0 | 0.8 | 0.0 | 0.0 | 0.2 | 0.0 | 0.2 |
| 7 to 10% | 0.0 | 0.8 | 0.7 | 0.0 | 0.5 | 0.0 | 0.4 |
| 4 to 6% | 1.3 | 1.5 | 1.4 | 4.1 | 1.8 | 1.5 | 1.7 |
| 1 to 3% | 28.8 | 26.1 | 24.5 | 25.7 | 26.0 | 25.8 | 26.3 |
| <1% | 69.9 | 70.9 | 73.5 | 70.3 | 71.5 | 72.7 | 71.5 |
| Grass tetany | n = 151 | n = 131 | n = 140 | n = 71 | n = 425 | n = 65 | n = 507 |
| >10% | 0.7 | 0.8 | 0.0 | 1.4 | 0.5 | 1.5 | 0.6 |
| 7 to 10% | 1.3 | 0.0 | 0.0 | 0.0 | 0.2 | 1.5 | 0.4 |
| 4 to 6% | 0.7 | 2.3 | 1.4 | 5.6 | 2.1 | 0.0 | 2.0 |
| 1 to 3% | 11.9 | 13.0 | 16.4 | 15.5 | 14.6 | 10.8 | 14.4 |
| <1% | 85.4 | 84.0 | 82.1 | 77.5 | 82.6 | 86.2 | 82.6 |
| Ketosis | n = 149 | n = 130 | n = 141 | n = 72 | n = 425 | n = 66 | n = 506 |
| >10% | 0.0 | 0.8 | 0.0 | 0.0 | 0.2 | 0.0 | 0.2 |
| 7 to 10% | 0.7 | 0.0 | 0.0 | 0.0 | 0.0 | 1.5 | 0.2 |
| 4 to 6% | 1.3 | 3.1 | 3.6 | 5.6 | 3.3 | 1.5 | 3.0 |
| 1 to 3% | 25.5 | 20.0 | 25.5 | 25.0 | 22.6 | 33.3 | 23.9 |
| <1% | 72.5 | 76.2 | 70.9 | 69.4 | 73.9 | 63.6 | 72.7 |

[a]Herds were categorized by herd size (large: >150 cows, above average: 100–150 cows, average: 60–100 cows, or small: <60 cows) based on the Irish national dairy herd average size (93 cows; [16]), and by calving pattern (spring-calving: cows calving in spring, or split-calving: cows calving in spring and autumn).

## Disease records

Disease incidence records were kept by <55.0% of respondents for any of the evaluated conditions (Table 2). The odds of farmers from small herds reporting to keep records of metritis incidence were lower than those of farmers from large herds (OR [95% CI] = 0.35 [0.14–0.83]; $P$ = 0.010; Table 3). No evidence of differing odds for reporting keeping incidence records for other conditions among farmers from different herd sizes and calving patterns was observed.

**Table 2. Reported dairy cow peripartum condition records kept (% of respondents to a transition period survey in Ireland).**

| Condition and record type | Herd size[a] | | | | Herd calving pattern[a] | | |
|---|---|---|---|---|---|---|---|
| | Large | Above average | Average | Small | Spring-calving | Split-calving | All |
| Milk fever | n = 153 | n = 131 | n = 145 | n = 74 | n = 430 | n = 66 | n = 518 |
| Antibiotic treatment | 17.6 | 24.4 | 22.1 | 40.5 | 25.1 | 18.2 | 23.7 |
| Supportive treatment | 30.7 | 29.0 | 35.9 | 35.1 | 33.5 | 27.3 | 32.6 |
| Incidence | 41.2 | 42.7 | 42.8 | 35.1 | 43.7 | 34.8 | 41.9 |
| Retained placenta (held cleaning) | n = 149 | n = 133 | n = 146 | n = 73 | n = 430 | n = 65 | n = 516 |
| Antibiotic treatment | 48.3 | 48.1 | 55.5 | 46.6 | 49.5 | 55.4 | 50.0 |
| Supportive treatment | 21.5 | 18.8 | 22.6 | 23.3 | 21.4 | 23.1 | 21.3 |
| Incidence | 49.7 | 49.6 | 43.8 | 32.9 | 47.9 | 43.1 | 45.9 |
| Metritis | n = 148 | n = 130 | n = 141 | n = 72 | n = 422 | n = 64 | n = 505 |
| Antibiotic treatment | 39.2 | 39.2 | 39.0 | 40.3 | 39.1 | 45.3 | 39.4 |
| Supportive treatment | 14.9 | 13.8 | 13.5 | 13.9 | 13.0 | 17.2 | 13.9 |
| Incidence | 41.9 | 30.8 | 31.2 | 19.4 | 35.5 | 28.1 | 33.7 |
| Displaced abomasum | n = 152 | n = 133 | n = 145 | n = 74 | n = 431 | n = 66 | n = 519 |
| Antibiotic treatment | 58.6 | 53.4 | 53.1 | 55.4 | 55.2 | 57.6 | 54.9 |
| Supportive treatment | 9.9 | 7.5 | 12.4 | 10.8 | 10.0 | 10.6 | 10.0 |
| Incidence | 36.8 | 33.8 | 35.2 | 31.1 | 36.2 | 34.8 | 35.1 |
| Grass tetany | n = 148 | n = 131 | n = 145 | n = 71 | n = 423 | n = 65 | n = 509 |
| Antibiotic treatment | 12.2 | 14.5 | 17.2 | 21.1 | 15.4 | 15.4 | 15.5 |
| Supportive treatment | 19.6 | 21.4 | 20.0 | 14.1 | 20.6 | 15.4 | 19.6 |
| Incidence | 31.8 | 34.4 | 28.3 | 21.1 | 32.2 | 21.5 | 30.3 |
| Ketosis | n = 150 | n = 124 | n = 134 | n = 73 | n = 414 | n = 64 | n = 498 |
| Antibiotic treatment | 22.7 | 22.6 | 22.4 | 31.5 | 22.9 | 26.6 | 23.5 |
| Supportive treatment | 18.7 | 15.3 | 16.4 | 13.7 | 15.5 | 18.8 | 16.3 |
| Incidence | 26.7 | 25.0 | 27.6 | 24.7 | 27.8 | 20.3 | 26.5 |

[a]Herds were categorized by herd size (large: >150 cows, above average: 100–150 cows, average: 60–100 cows, or small: <60 cows) based on the Irish national dairy herd average size (93 cows; [16]), and by calving pattern (spring-calving: cows calving in spring, or split-calving: cows calving in spring and autumn).

Farmers frequently reported keeping records of antibiotic treatments for displaced abomasum and/or digestive problems (54.9%; 285/519), retained placenta (held cleaning; 50.0% [258/516]), and metritis (39.4% [199/505]; Table 2). Additionally, some farmers, reported keeping records of antibiotic treatments for metabolic conditions (i.e. milk fever [23.7%; 123/518], ketosis [23.5%; 117/498] and grass tetany [15.5%; 79/509]; Table 2). The odds of farmers from small herds reporting keeping records of antibiotic treatments for milk fever were over 3 times those of farmers from large herds (OR [95% CI] = 3.2 [1.42–7.26]; $P < 0.001$), while no evidence of differing odds between farmers from average and above average compared to those of farmers from large herds was observed (Table 3).

## Dry cow management

Reported management strategies by herd size and calving pattern is presented in S5 Table. Most commonly implemented management strategies for dry cows were body condition monitoring (73.4%; 365/497) and Mg and/or dry cow mineral supplementation in diet (61.2% [304/497]; Fig 3). The least reported management strategies were feeding a low K diet (20.3%; 101/497) or an acidifying diet (dietary cation-anion difference [DCAD]; 6.2% [31/497]; Fig 3). Some differences between reportedly implemented management strategies by herd size and calving pattern were observed (Table 3; Fig 3). Managing dry cows in more than one group

**Table 3. Herd size odds ratios and 95% CI for responses to questions from a transition period survey in Ireland.**

| Survey question and answer[a] | Class contrast (Herd size/calving pattern)[a] | Odds ratio (95% CI)[b] | P-value[c] |
|---|---|---|---|
| Management strategy | | | |
| Management in >1 group | Small vs. Large | 0.22 (0.10, 0.50) | <0.001 |
| | Average vs. Large | 0.73 (0.40, 1.33) | 0.537 |
| | Above average vs. Large | 0.64 (0.40, 1.03) | 0.249 |
| Management in >1 group | Split- vs. spring-calving | 0.51 (0.30, 0.86) | 0.011 |
| Provide feed sources except silage | Split- vs. spring-calving | 2.48 (1.46, 4.24) | <0.001 |
| Once-a-day milking after calving | Split- vs. spring-calving | 0.16 (0.07, 0.38) | <0.001 |
| Cows indoors for a period after calving | Split- vs. spring-calving | 0.34 (0.20, 0.57) | <0.001 |
| Disease treatment incidence | | | |
| Milk fever | Split- vs. spring-calving | 1.78 (1.02, 3.12) | 0.042 |
| Record type | | | |
| Metritis incidence | Small vs. Large | 0.35 (0.14, 0.83) | 0.010 |
| | Average vs. Large | 0.63 (0.34, 1.17) | 0.223 |
| | Above average vs. Large | 0.63 (0.33, 1.20) | 0.256 |
| Antibiotic usage for milk fever | Small vs. Large | 3.20 (1.42, 7.26) | <0.001 |
| | Average vs. Large | 1.30 (0.61, 2.74) | 0.808 |
| | Above average vs. Large | 1.48 (0.69, 3.13) | 0.546 |

[a]Herds were categorized by herd size (large: >150 cows, above average: 100–150 cows, average: 60–100 cows, or small: <60 cows) based on the Irish national dairy herd average size (93 cows; [14]), and by calving pattern (spring-calving: cows calving in spring, or split-calving: cows calving in spring and autumn).
[b]Contrast analysed as "yes" vs. "no" except for milk fever reported treatment incidence (≤3% or >3%).
[c]Values were adjusted using the Tukey-Kramer adjustment for multiple comparisons in the herd size model.

(e.g. separate groups for fat and thin cows) was less frequently reported by farmers from small than large herds (OR [95% CI] = 0.22 [0.10–0.50]; $P < 0.001$) and by farmers from split- than spring-calving herds (OR [95% CI] = 0.51 [0.30–0.86]; $P = 0.011$; Table 3). The odds of farmers from split-calving herds reporting the provision of feeds other than silage to dry cows were 2.5 times those of farmers from spring-calving herds (OR [95% CI] = 2.48 [1.46–4.24]; $P < 0.001$; Table 3).

## Fresh cow management

Reported management strategies by herd size and calving pattern is presented in S6 Table. The most commonly implemented fresh cow management strategy in relation to transition cow disease prevention was Ca supplementation at calving (60.6% [314/487]; Fig 4); of these, 82.2% (258/314) reported supplementing only "high-risk" cows and 12.1% (38/314) reported supplementing all cows (18 respondents chose both options). Some differences in implemented management strategies by herd size and calving pattern were observed (Table 3; Fig 4). Milking cows once-a-day for a period after calving was less frequently reported by farmers from split- calving than from spring-calving herds (OR [95% CI] = 0.16 [0.07–0.38]; $P < 0.001$). Last, keeping freshly calved cows indoors for a period after calving (the overall most frequently reported management strategy; 68.0% [331/487]) was less frequently reported by farmers from split- than spring-calving herds (OR [95% CI] = 0.34 [0.20–0.57]; $P < 0.001$).

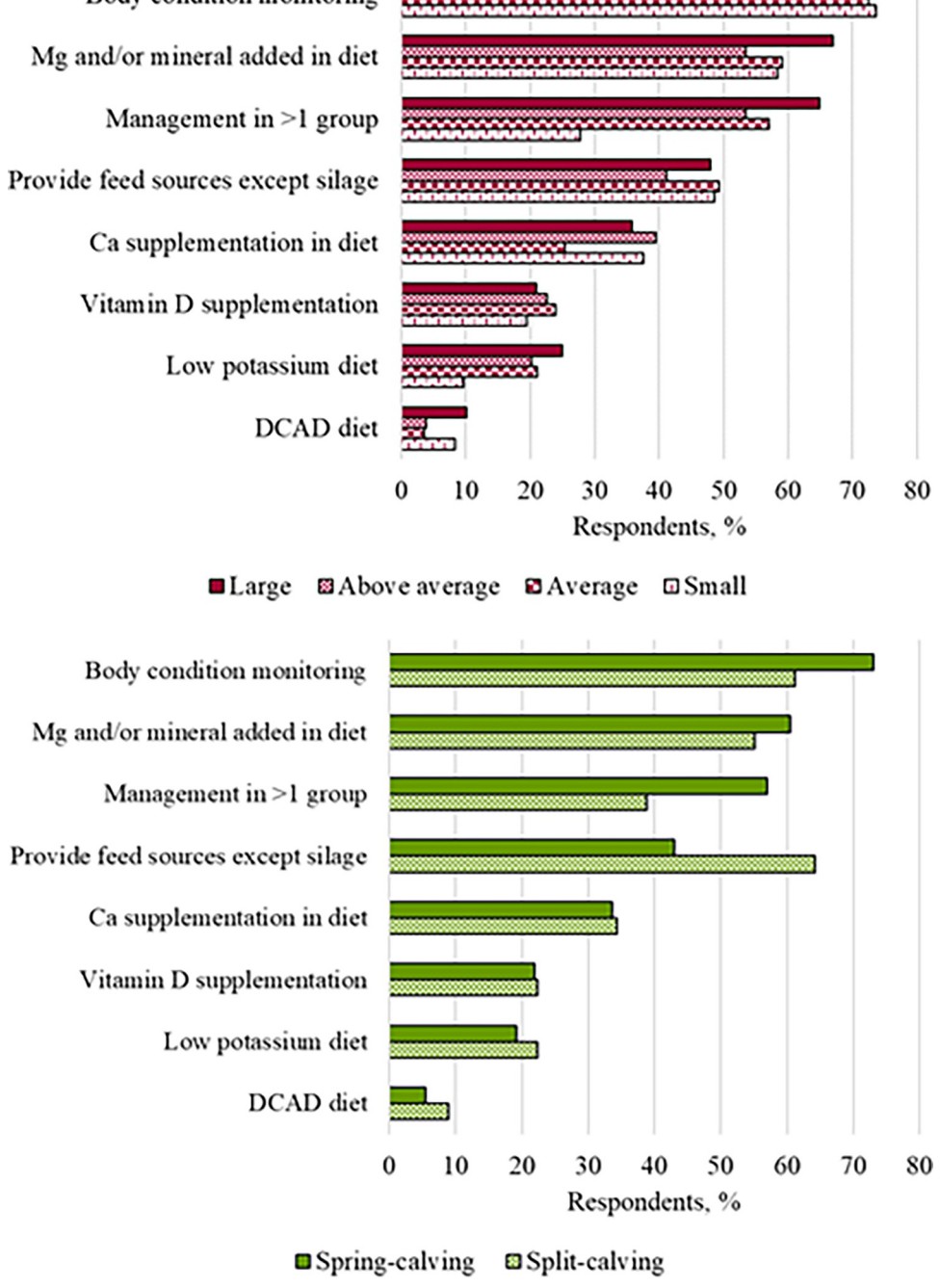

**Fig 3.** Reported dry cow management strategies implemented, by herd size (A; large [>150 cows; n = 148], above average [100–150 cows; n = 129], average [60–100 cows; n = 142] and small [<60 cows; n = 72]) and herd calving pattern (B; spring-calving [n = 428] and split-calving [n = 67]) for respondents to a transition period survey in Ireland.

## Discussion

A final total of 525 responses were suitable for data analysis, this represents 3.4% of Irish dairy herds (total of 15,319 dairy herds in 2022; [16]). Overall, respondents to this survey had larger herds and above average performance when compared to national averages; respondents

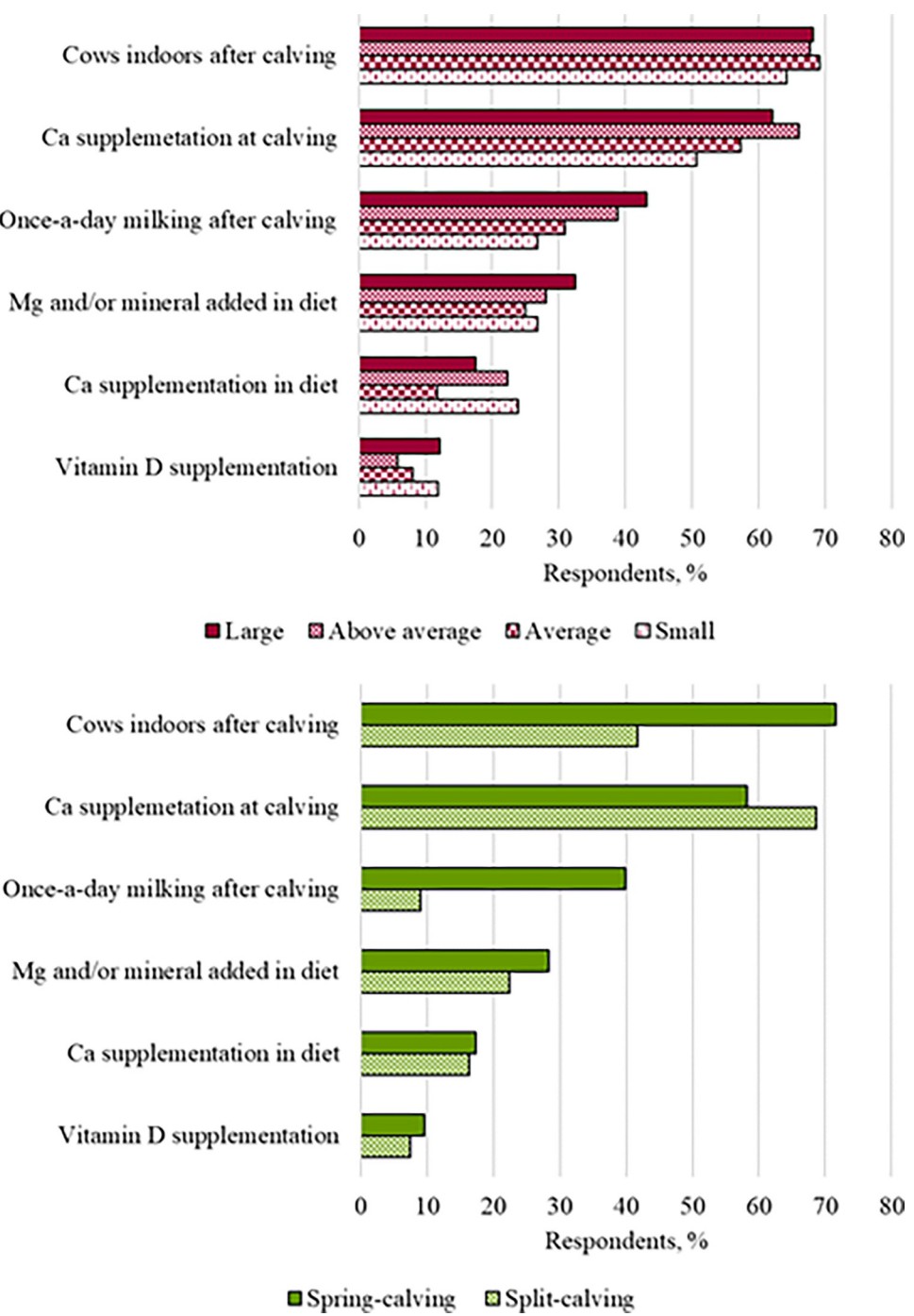

**Fig 4.** Reported fresh cow management strategies implemented, by herd size (A: large [>150 cows; n = 148], above average [100–150 cows; n = 121], average [60–100 cows; n = 136] and small [<60 cows; n = 67]) and herd calving pattern (B: spring-calving [n = 416] and split-calving [n = 67]) for respondents to a transition period survey in Ireland.

mean herd size was 45% higher than the mean dairy herd size in the Republic of Ireland which is 93 cows [9], respondents mean 305-day milk yield and calving interval were respectively 20% higher and 3% lower than the 2022 national means (5,716 L/cow [11] and 388 days [ICBF HerdPlus users] [11]). The apparent 'above average' profile of the respondents' herds is not

surprising as this survey was distributed among Teagasc Technical Dairy Advisory clients which tend to operate at a higher standard of technical and financial performance than the overall dairy farmer population in Ireland [13]. It has to be noted that farmers chose to fill in the survey, thus further potential bias exists in the sample population as farmers interested in, or who are experiencing some issues with transition cow health and management may have been more likely to answer this survey. Recognising that the definition for the transition period provided in this survey (i.e. "late dry [late pregnancy if primiparous] to early lactation period") along with some other phrases used throughout the survey (e.g. "freshly calved", "early cal-vers"), are subjective, respondents may have perceived the timelines in which the questions were asking about differently which should be considered when interpreting results.

Respondents mostly had spring-calving herds which are most commonly seen in Ireland given the seasonal grass growth (92% of dairy herds; [14]) and the majority of respondents were located in county Cork (36.0% [147/408]; Fig 1), which is the county with the highest number of dairy cows in Ireland [15]. Regarding the reported disease levels, a herd alarm milk fever incidence threshold of >3% (within 14 days post-calving) was described by Lean and DeGaris [12] in an Australian technical review using data from grazing and confined herds; based on this threshold, 23.0% of respondents to our survey should be seeking help in regards to milk fever prevention. The provided threshold for retained placenta in this same review (>12 hours after calving; >6%) suggests that 2.5% of respondents to our survey should be seeking help for this condition if their definition of retained placenta aligned with the one used in the review, however, no definition of retained placenta was provided in the survey and respondents may have assumed varying definitions. Subclinical hypocalcaemia is a recurrent topic of research worldwide as reviewed by Couto Serrenho et al. [16] suggesting that transfer (or uptake) of scientific outputs to Irish dairy farmers may be limited (21% reported not know-ing if subclinical hypocalcaemia was a problem in their herd).

Given the low number of farmers reporting to keep disease records, the creation and promo-tion of strategies to improve record-keeping on farms should be an area of focus for outreach activities. Disease incidence and treatment record-keeping is paramount in identifying patterns of disease and in aiding management of a disease at herd-level [17]. Our results also suggest that inappropriate antibiotic treatment decisions for metabolic disease treatment may be made at the farm-level. In the context of confined cows where extra-label use of antibiotics in the peri-partum has been described, training the farmworkers involved in administering treatments to sick cows has proved successful at increasing their knowledge on transition cow disease diagno-sis and treatment, without succeeding at decreasing overall antimicrobial use on farm [18, 19].

In terms of dry cow management strategies, the importance of optimizing body condition at calving for subsequent health and reproductive performance and Mg supplementation to reduce the risk of milk fever in grazing systems has been emphasized for decades, thus, it is not surprising that the message has reached Irish dairy farmers and these are commonly reported dry cow management strategies [20–22]. In agreement with our findings, an Irish survey by Cummins et al. [15] reported that most of the respondents to their survey (n = 262) set a target calving BCS and fed dry cow minerals. Managing cows in >1 group during the dry period was one of the most commonly reported management strategies for this period, grouping cows by BCS is recommended for optimal BCS management during the dry period and BCS monitor-ing was the most commonly reported strategy in this study, however, we did not enquire about the management associated with the grouping strategy.

Low K diets are recommended for transition cows given K's contribution to a positive DCAD ultimately interfering with calcium metabolism and impairing dietary Mg absorption. Negative DCAD diets have solidly proven successful for milk fever prevention in confined cows [23, 24]. The high K concentration and DCAD in pasture have been described as limiting

factors for the implementation of these strategies in grazing systems [25]. Research in grazing cows reports no association between positive DCAD (350 to 535 mEq/kg DM) and high K concentration (3.3 to 4.2% of DM) in pasture and plasma Ca concentration at calving, suggesting that these potential determinants of milk fever risk may not be as important in grazing dairy systems as they are in confined systems [26]. Nevertheless, K concentrations in Irish grass silage may not be as high as those reported from pasture in New Zealand studies (mean [range] = 2.4% [0.6 to 5.6%] of DM; n = 1,636 samples; [27]); and thus opting for a low K grass silage or achieving a lower DCAD through the addition of anionic salts may be management strategies more suitable for dry cow feeding in the Irish dairy production system than in other grazing systems. Therefore, further research is needed to understand the limited uptake and to identify the barriers for the adoption and implementation of these research-supported strategies for milk fever prevention by Irish dairy farmers.

The most reported fresh cow strategy was keeping cows indoors for a period postpartum, a practice more commonly implemented in spring-calving dairy herds. This strategy is most likely implemented due to excessive soil moisture during the first months of the spring calving season (January and February; [28]) rather than by a transition cow health improvement desire. Split-calving herds use a lower amount of grazed grass in their cows diet potentially explaining the lower implementation among these farmers [29]. Calcium supplementation at calving was the next most commonly reported strategy that could be associated with a transition cow health improvement desire; this practice is regarded as a prophylactic strategy for hypocalcaemia, effective at temporarily increasing blood Ca concentration and leading to positive performance effects on subpopulations of animals [30, 31]. Within this survey question, answers of supplementing "high-risk cows" and supplementing "all cows" at calving were combined; we did not ask farmers to outline their definition of a "high-risk" cow or their supplementation protocol, both of which are paramount in reaping the benefits of this management strategy according to research conducted in confined cows. To the best of our knowledge, only two studies by the same authors have evaluated Ca supplementation at calving in commercial Irish dairy farms [30, 32]; these studies assessed the safety and efficacy of a calcium and antioxidant bolus and focussed on the metabolic status and milk production of the cow post supplementation.Further research evaluating Ca supplementation strategies in the Irish dairy production context is warranted to optimise this commonly implemented strategy. Once-a-day milking was the third most popularly reported fresh period management strategy, this practice enables labour savings [33], and may reduce metabolite imbalances in early lactation and also reduce days to conception after calving [34–36]; nevertheless our study did not enquire about the reasons behind the reported management strategies.

## Conclusions

Results from the present study suggest that milk fever is a transition cow health concern in Irish dairy farms. Optimization of commonly implemented dry cow (Mg and/or dry cow mineral supplementation) and fresh cow (Ca supplementation at calving) management strategies, as well as enhanced uptake of dry cow management strategies proven successful under other production systems (low K and negative DCAD diet) may help reduce milk fever's burden on Irish dairy farms. Further research should identify the factors limiting the effectiveness of implemented management strategies and the end user adoption of successful management strategies for milk fever prevention. Additionally, dissemination activities targeting farmers from all herd sizes would be beneficial to increase awareness of peripartum metabolic diseases and their recommended treatment, as well as to promote disease incidence and treatment record keeping.

## Supporting information

**S1 Table. Transition cow health and management questions sent in a survey via text message to 3 899 Teagasc (Agriculture and Food Development Authority in the Republic of Ireland) dairy advisory clients in October 2022.** The survey was made using Survey Monkey (SurveyMonkey Inc., Palo Alto, CA) but has been presented here in table format.
(DOCX)

**S2 Table. Dairy cow herd descriptions for survey respondents with an active ICBF (Irish Cattle Breeding Federation) account by herd size and calving pattern.** [a]Herds were categorized by herd size (large: >150 cows, above average: 100–150 cows, average: 60–100 cows, or small: <60 cows) using the Irish national dairy herd average as reference (93 cows; [9]), and by calving pattern (spring-calving: cows calving in spring, or split-calving: cows calving in spring and autumn). [b]IQR = Interquartile range.
(DOCX)

**S3 Table. Reported highest observed disease incidence by cow parity, stage of lactation and stage of calving season presented by herd size and calving pattern (% of respondents to a transition period survey in Ireland).** [a]Herds were categorized by herd size (large: >150 cows, above average: 100–150 cows, average: 60–100 cows, or small: <60 cows) using the Irish national dairy herd average as reference (93 cows; [9]), and by calving pattern (spring-calving: cows calving in spring, or split-calving: cows calving in spring and autumn). [b]Stages of lactation: Fresh calver: First 3 weeks after calving, early lactation: from week 3 to end of $3^{rd}$ month of lactation, mid lactation: from start of $4^{th}$ month to end of $7^{th}$ month of lactation, late lactation: from start of $8^{th}$ month of lactation to dry-off, far-off dry: from dry-off to close-up, close-up dry: last 3 weeks of pregnancy.
(DOCX)

**S4 Table. Reported perception of dairy cow diseases by herd size and calving pattern.** Perception was based on treatments, mortality, culling and herd performance (% of respondents to a transition period survey in Ireland). [a]Herds were categorized by herd size (large: >150 cows, above average: 100–150 cows, average: 60–100 cows, or small: <60 cows) using the Irish national dairy herd average as reference (93 cows; [9]), and by calving pattern (spring-calving: cows calving in spring, or split-calving: cows calving in spring and autumn). [b]Perception definitions: Significant problem (regularly treating severe cases with some cows lost/culled), routine problem (regularly treating cows to control issues), occasional cases (but no major effect on herd performance).
(DOCX)

**S5 Table. Reported dry cow management strategies by herd size and calving pattern (% of respondents to a transition period survey in Ireland).** [a]Herds were categorized by herd size (large: >150 cows, above average: 100–150 cows, average: 60–100 cows, or small: <60 cows) using the Irish national dairy herd average as reference (93 cows; [9]), and by calving pattern (spring-calving: cows calving in spring, or split-calving: cows calving in spring and autumn). [b]DCAD = Dietary cation anion difference.
(DOCX)

**S6 Table. Reported fresh cow management strategies by herd size and calving pattern (% of respondents to a transition period survey in Ireland).** [a]Herds were categorized by herd size (large: >150 cows, above average: 100–150 cows, average: 60–100 cows, or small: <60 cows) using the Irish national dairy herd average as reference (93 cows; [9]), and by calving pattern (spring-calving: cows calving in spring, or split-calving: cows calving in spring and

autumn).
(DOCX)

## Acknowledgments

The authors are especially thankful to the farmers who responded to this survey. And to J. Mason (Teagasc, Animal and Grassland Research and Innovation Centre, Moorepark, Fermoy, Ireland) for her work compiling herd-level information. Financial support from the Irish Dairy Levy (Dairy Research Ireland, Dublin, Ireland) and Teagasc Walsh scholarship programme is greatly acknowledged.

## Author Contributions

**Conceptualization:** Ainhoa Valldecabres.

**Data curation:** Louise Horan, Ainhoa Valldecabres.

**Formal analysis:** Louise Horan.

**Funding acquisition:** John F. Mee, Ainhoa Valldecabres.

**Investigation:** Louise Horan, Ainhoa Valldecabres.

**Methodology:** Louise Horan, Joseph Patton, Conor G. McAloon, John F. Mee, Ainhoa Valldecabres.

**Project administration:** Ainhoa Valldecabres.

**Supervision:** Ainhoa Valldecabres.

**Writing – original draft:** Louise Horan.

**Writing – review & editing:** Joseph Patton, Conor G. McAloon, Ángel García-Muñoz, Áine Regan, John F. Mee, Ainhoa Valldecabres.

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
