## [Decision Letter · Decision Letter 0]

27 Sep 2024

PONE-D-24-31981Transition cow health and management in pasture-based dairy herds: a farmers’ surveyPLOS ONE

Dear Dr. Valldecabres,

Thank you for submitting your manuscript to PLOS ONE. After careful consideration, we feel that it has merit but does not fully meet PLOS ONE’s publication criteria as it currently stands. Therefore, we invite you to submit a revised version of the manuscript that addresses the points raised during the review process. As you can see below, the reviewers showed contrasting opinions about the merit of the manuscript. In light of the split recommendations, I invite you to carefully address all the reviewers' comments. Please pay careful attention to the criticism from reviewer #3. They raise valid points that must be considered when revising the manuscript. Also, this invitation to revise the manuscript is no guarantee for the manuscript's acceptance but rather an opportunity to revise and improve the manuscript to overcome the shortcomings identified.

We look forward to receiving your revised manuscript.

Kind regards,

Angel Abuelo, DVM, MRes, MSc, PhD, DABVP (Dairy), DECBHM

Academic Editor

PLOS ONE

Journal Requirements:

2. We note that you have referenced (Reardon et al., unpublished) which has currently not yet been accepted for publication. Please remove this from your References and amend this to state in the body of your manuscript: (ie “Bewick et al. [Unpublished]”) as detailed online in our guide for authors

3. We note that [Figure 1] in your submission contain [map/satellite] images which may be copyrighted. All PLOS content is published under the Creative Commons Attribution License (CC BY 4.0), which means that the manuscript, images, and Supporting Information files will be freely available online, and any third party is permitted to access, download, copy, distribute, and use these materials in any way, even commercially, with proper attribution. For these reasons, we cannot publish previously copyrighted maps or satellite images created using proprietary data, such as Google software (Google Maps, Street View, and Earth). For more information, see our copyright guidelines: http://journals.plos.org/plosone/s/licenses-and-copyright.

Reviewers' comments:

Reviewer's Responses to Questions

**Comments to the Author**

1. Is the manuscript technically sound, and do the data support the conclusions?

Reviewer #1: Yes

Reviewer #2: Yes

Reviewer #3: No

2. Has the statistical analysis been performed appropriately and rigorously? 

Reviewer #1: Yes

Reviewer #2: Yes

Reviewer #3: No

3. Have the authors made all data underlying the findings in their manuscript fully available?

Reviewer #1: Yes

Reviewer #2: No

Reviewer #3: No

4. Is the manuscript presented in an intelligible fashion and written in standard English?

Reviewer #1: No

Reviewer #2: Yes

Reviewer #3: Yes

5. Review Comments to the Author

Reviewer #1: Thank you for your paper - I have minor comments on attached annotated PDF. I think you need to get all co-authors to read through and check for grammatical errors as although I have pointed some out there are more

Reviewer #2: The manuscript is well presented and would be of great interest to producers in Ireland. I have a few comments below.

I could not find a data availability statement. Please include per PLOS ONE requirements.

Line 200-205: It is unclear as to what the different categories are when only highlighting the category with the highest percentage. I suggest moving ln 206-207 to the start of this paragraph so the reader can reference that table as they progress through the paragraph. Likewise, moving ln. 211-212 prior to Line 208 (Overall most farmers…). I recommend this change throughout the manuscript.

Line 330: Do farmers not know what subclininical hypocalcemia is or do they not know if it is a problem since subclinical hypocalcemia does not have clinical signs, making it difficult to diagnose? In addition, how were the questions for Table S4 posed in the survey? If a farm does not record a disorder/monitor it, then they may respond with “I don’t know” versus one of the other options. Therefore, could there have been misinterpretation with the questions/options such that “I don’t know” was not interpreted as them not knowing what the disorder was? These comments are also applicable to subclinical ketosis - though subclinical ketosis is easier to monitor on farm, if the farm regularly monitors BHB. Blood calcium analysis is typically more limited and I would say farmers likely do not routinely monitor blood Ca to determine if the cow has subclinical hypocalcemia.

I suggest including the farm location (i.e. Ireland) and how this data was obtained (i.e. a survey) in your table and figure captions.

Reviewer #3: Transition cow management in pasture-based cows: a farmer survey.

The authors undertake a qualitative survey of farmer perceptions towards the importance of the transition period to health and productivity. I’m a strong supporter of consulting end users on their needs/opportunities and of proactively using data to develop hypotheses to test. So, the authors deserve credit for their initiative and endeavors. The authors conclude that, because of their survey results, more research is needed on milk fever/hypocalcemia, in particular, retained placenta, BCS monitoring, peripartum housing facilities, and early lactation calcium supplementation. This is despite the century of research that has already been undertaken in these topic areas. As someone who has worked extensively in this area for a long time, I do not agree with these conclusions and the authors offer no basis for the conclusion other than these diseases/poor metabolic adaptations are regarded as important by Irish dairy farmers. I remain to be convinced that this work reflects the ‘Irish situation’. Response rate is low (although an appreciable number of farmers) and very likely to be biased to farmers who have trouble with peripartum metabolic disease. The manuscript is very long, with very few results of note extensively discussed in the form of a type of literature review to support the authors’ position.

The Discussion subsections are long – excessively so - more akin to a literature review than the discussion of your results. A combined ‘Results and Discussion’ section, would help address this, with some discipline to shorter passages. The literature is cited relies too heavily on reviews/summary works and book chapters – the authors need to look at the original experiments published.

The literature cited is primarily from housed/mixed ration systems and ignores a large body of pasture-based (where cows are fed pasture through the transition period) research. To be fair, the authors do acknowledge some pasture-based work by Roche and colleagues in NZ. But, here they mainly cite the reviews and book chapters (some of which are behind paywalls, unfortunately), and not the large body of literature amassed over the last 20-30 years by this same team led by Roche, much of which provides some answers to many of the questions being raised by these Irish farmers. Instead, the majority of the discussion supporting the authors’ assertions comes from TMR systems.

In conclusion, in my opinion, it is important to survey farmers as you’ve done. However, this dataset is limited in its usefulness. -I do not agree that your results support further research in the areas you’ve highlighted. They might support greater advisory effort in these areas, but that is a very different conclusion and you would have to review the advisory material and how it is delivered to determine if this is true and the best solutions. -I believe the contents of the manuscript, if shortened to focus on the most important findings, would be an appropriate conference topic, perhaps, or if diligence to brevity in scientific writing could be pursued, perhaps it could be acceptable as a Brief Communication if this journal accepted this journal.

6. PLOS authors have the option to publish the peer review history of their article (what does this mean?). If published, this will include your full peer review and any attached files.

Reviewer #1: No

Reviewer #2: No

Reviewer #3: No

---

## [Author Response · Author response to Decision Letter 0]

11 Nov 2024

We sincerely thank the reviewers for their time and input to improve our manuscript; below we provide specific answers to each comment along with captions from the text to facilitate the review process. Changes to the manuscript have been highlighted in yellow.

Reviewer #1

Abstract

1. Most pasture-based systems use housing - grass in summer, housing in winter. All pasture all the time systems are the unusual ones.

AU: Accepted. Thank you for this comment. We tried to differentiate Ireland’s from New Zealand’s production as most of the research in transition pasture-based cows comes from that region. We acknowledge the reviewer’s comment and have edited accordingly (Removed the phrase “Compared to other grazing systems,” and now the sentence reads “During the dry period, cows are housed and offered predominantly grass silage, providing unique transition cow management opportunities.” [Lines 25-26]).

2. % of what - please make clear and report denominators. how can you have three 'highest' disease incidence periods

AU: Accepted. Thank you for this comment. Denominators are now reported throughout the abstract. These numbers correspond to answers to three different questions: Q1, Q2 and Q3 (see Table S1). We have changed the phrasing (lines 33 – 35: “Disease incidence was reportedly highest in cows within their first 3 weeks postpartum (58%; 299/519), in cows calving at the end of the block calving season (48%; 245/510) and in multiparous cows (52%; 266/513).”)

3. Here and elsewhere - it's not 'reported to treat' it's 'reported treating' or similar

AU: Accepted. Thank you for this comment. The phrasing of relevant sentences has been changed accordingly throughout the manuscript.

4. What does this group mean - aren't study herds all spring calving herds. If not make it clear they're not - and state proportions

AU: Accepted. Thank you for this comment. We have now included a brief description of study participants herds calving patterns at the beginning of the abstract for clarity (lines 31 – 33: “Results are presented for all respondents, by herd size and by the two most common calving systems (spring- [84.3%; 439/521] and split-calving [12.9%; 67/521]).”)

5. Reporting the provision of feeds - again it's not 'to'

AU: Accepted. Thank you again for this comment, it has now been changed accordingly throughout the manuscript.

6. Why is anybody using antibiotics to treat milk fever cases?

AU: Farmers reported to keep records of antibiotic treatments for milk fever, it implies that antibiotics were used to treat milk fever, but we did not ask specifically for it or their reasons for this practice. Perhaps something that could be confronted in future studies regarding responsible antibiotic usage.

7. I don't think this is a surprising finding but it's presented as such

AU: Accepted. Thank you for this comment. We have changed the phrasing of the conclusion part of the abstract to “In conclusion, responses to our survey suggest…” (line 44). 

Introduction

8. Get the co-authors to check the English

AU: Accepted. Thank you for this comment. The English has been reviewed throughout the manuscript.

9. Actually much of the transition period is actually homeorhesis - see Bradford and Shwartz 2020 and Degaris and Lean 2008

AU: Accepted. Thank you for this comment, it was very helpful. Changes to reflect that that homeorhesis is challenged during the transition period have been made accordingly to this sentence and Bauman and Currie (1980) has now been referenced (lines 50 – 53: “This is not surprising given the range of physical (physiological, immunological and metabolic) and environmental changes which challenge cows’ homeostasis and homeorhesis, often turning into disease and ultimately impairing cows’ welfare and production performance [1].”).

10. Please rephrase as I had to read this multiple times to work out what it mean - perhaps start with Roche [8} interviewed ... and found ...

AU: Accepted. Thank you for this comment. The sentence has been rephrased and we noticed an error with the reference which has been changed (lines 59 – 62: “Redfern et al. [3] interviewed farm advisors and reported that advisors were not providing farmers with focussed advise due to time constraints and fear of responsibility, among others. This lack of focussed advice being given to farmers may also restrain the improvement on transition cow health and management.”).

11. In the abstract this was a bit vague but this is now more explicitly wrong - many systems use grazing and housing - Ireland's uniqueness is the combination ina system where grass (grazed or preserved is by far the most important feed)

AU: Accepted. Thank you for this comment. Changes have been made accordingly to highlight the proportion of grass in the dairy cows’ diet as the unique factor that Irish dairy production has over other pasture-based systems (lines 69 – 71: “Forage, mostly grazed pasture, makes up 95% of the Irish dairy cows’ diet [5], creating a need for bespoke transition cow management.”).

12. Definitely needs bespoke transition management but it is far from unique

AU: Accepted. Thank you for this comment. We have changed the phrasing of this sentence to lead the reader to understand the requirement for bespoke over “unique” management. Please see comment 11 above. 

13. Doesn't Roche's AVS supplement paper - The incidence and control of hypocalcaemia in pasture-based systems. have Irish data

AU: The authors could not access the currently 21 years old paper mentioned which is published under a non-open access policy. While Roche may have published using Irish data, we refer to a lack of national-level data (i.e. lack of studies such as the NAHMS survey in the US or the recent work by Kerwin and colleagues in Cornell).

Materials and methods

14. this makes it sound like they became advisors

AU: Accepted. Thank you for this comment, this has now been changed to make it sound less like this (line 95: “Irish farmers voluntarily sign up to the advisory service…”)

15. clients

AU: Accepted. Thank you for noticing this, it has now been changed from members to clients (line 99). 

16. what does late dry mean? Last 30 days? last 7 days? What do you think yur respondents thought it meant? Same applies to early lactation

AU: Late dry and early lactation were not defined at the beginning of this survey and this is now highlighted as a study limitation in the discussion (lines 318 – 323: “Recognising that the definition for the transition period provided in this survey (i.e. “late dry [late pregnancy if primiparous] to early lactation period”) along with some other phrases used throughout the survey (e.g. “freshly calved”, “early calvers”), are subjective, respondents may have perceived the timelines in which the questions were asking about differently which should be considered when interpreting results.”). 

17. repetition - use longer version first

AU: Accepted.

18. what does this mean Is survey available as supplementary material

AU: Accepted. Thank you for this comment. An explanation has been added to this line (lines 116 – 117: “The second question asked farmers for their herd number for the purpose of data extraction from the ICBF database …”). We have clarified why these questions are not available in the supplementary material “Given their lack of association to the survey results, these two questions are not included in the survey available as supplementary material (S1 Table).” (Lines 118 – 120).

19. participating (partaking in is more commonly used for food/drink)

AU: Accepted (line 118).

20. early lactation is your term

AU: We acknowledge that the reviewer recommendation is accurate, terms could have been chosen better, however it was not noticed designing the survey or during the validation process. Since in the original survey, we referred to “freshly calved” cows, the authors consider it more accurate to maintain the original terminology in the report. This kind of limitation is discussed in lines 318 – 323.

21. repetition

AU: Accepted. Thank you for bringing this to our attention. The repeated statement has now been deleted.

Results

22. equally affects surely otherwise the question is close to meaningless as

AU: Accepted. Thank you for this comment. This line has been changed to avoid confusion (line 210: “and that disease equally affects both, primiparous and multiparous cows”).

23. technically as the maternal placenta is always retained retained fetal membranes is a better term

AU: While we understand that retained fetal membrane would be a more biologically correct term here, “retained placenta (held cleaning)” was the term used in the survey questions. The phrase “(held cleaning) has now been added when referring to retained placenta throughout the manuscript to help clarify what is meant by this phrase. We consider it more accurate to maintain the consistency in terminology throughout the manuscript. In the discussion, the papers we reference for retained placenta herd alarm level also reference the condition as such. 

24. Herds

AU: Accepted. Thank you for this comment, the table caption has been changed accordingly (Table 1). 

25. vague term

AU: Accepted, the word “noticeable” has now been changed to “substantial” as the word substantial can be defined as something that is of considerable size, and this is what we are trying to communicate here (line 237). 

26. same question as abstract - what does this mean

AU: Please see author response to comment 6 above.

27. I can see why you focus on OR where 95%CI exclude 1, but if you're going to claim results are similar then you need CI to show that you have sufficient data to exclude the difference being quite large. Table 3 shows you don't

AU: Accepted, thank you for this comment. We have rephrased the references to non-statistically significant odds ratio. “No evidence of differing odds for reporting keeping incidence records for other conditions among farmers from different herd sizes and calving patterns was observed.” lines 247 – 249 and “while no evidence of differing odds between farmers from average and above average compared to those of farmers from large herds was observed (Table 3).” lines 257 - 258. 

Discussion

28. was that the definition your farmers used?

AU: No, this was not necessarily the definition farmers used for retained placenta as mentioned in lines 333 – 334 (“if their definition of retained placenta aligned with the one used in the review,”). However, to clarify that we did not provide farmers with a definition for retained placenta we have included the following at the end of this same sentence “however, no definition of retained placenta was provided in the survey and respondents may have assumed varying definitions.” lines 334 - 335.

29. 21

AU: Thank you for bringing this to our attention, however this reference has since been removed from the manuscript when addressing other reviewers’ comments. 

30. I'm not sure you can make this conclusion. the way I read the question is do you keep records of antibiotics given to cows with milk fever rather than do you use antibiotics to treat milk fever. The question is about recording not use

AU: Thank you for this comment. In the question we specify “in association with the following conditions”. Thus, we expect the farmer to have reported keeping records in association with the condition (i.e. antibiotics for milk fever). 

31. I think this is overstating it especially as Roche then says K is not unimportant

AU: Accepted. Thank you for this comment. This sentence has now been changed to more appropriately reflect what John Roche was saying in his paper (lines 367 – 368: “suggesting that these potential determinants of milk fever risk may not be as important in grazing dairy systems as they are in confined systems [26].”). 

Reviewer #2

1. I could not find a data availability statement. Please include per PLOS ONE requirements.

AU: Accepted. Thank you for your comment. Survey response details are confidential as our survey was not anonymous (Lines 128-129), and permission was only sought form the respondents for the purpose of this study and contact for other specific study. All the relevant results are summarized among the manuscript and the supplemental material.

2. Line 200-205: It is unclear as to what the different categories are when only highlighting the category with the highest percentage. I suggest moving ln 206-207 to the start of this paragraph so the reader can reference that table as they progress through the paragraph. Likewise, moving ln. 211-212 prior to Line 208 (Overall most farmers…). I recommend this change throughout the manuscript.

AU: Accepted. Thank you for this comment and your suggestion on how to remedy the problem. Changes have been made accordingly throughout the manuscript. 

3. Line 330: Do farmers not know what subclininical hypocalcemia is or do they not know if it is a problem since subclinical hypocalcemia does not have clinical signs, making it difficult to diagnose? In addition, how were the questions for Table S4 posed in the survey? If a farm does not record a disorder/monitor it, then they may respond with “I don’t know” versus one of the other options. Therefore, could there have been misinterpretation with the questions/options such that “I don’t know” was not interpreted as them not knowing what the disorder was? These comments are also applicable to subclinical ketosis - though subclinical ketosis is easier to monitor on farm, if the farm regularly monitors BHB. Blood calcium analysis is typically more limited and I would say farmers likely do not routinely monitor blood Ca to determine if the cow has subclinical hypocalcemia.

AU: Accepted. Thank you for this comment. The question relating to S4 Table can be seen in S1 Table question 4. We understand the potential misinterpretation that is outlined in this comment and have made two changes to the manuscript to help clarify that these respondents reported not knowing if the subclinical condition affects their herd rather than them reporting not knowing what it was (lines 256 – 241 – 242: “20.7% (107/517) of farmers reported not knowing if subclinical hypocalcaemia was a problem in their herd.” and lines 337 – 338: “…(21% reported not knowing if subclinical hypocalcaemia was a problem in their herd).”).

4. I suggest including the farm location (i.e. Ireland) and how this data was obtained (i.e. a survey) in your table and figure captions.

AU: Accepted. Thank you for this comment. The table and figure captions have been edited accordingly.

Reviewer #3

The authors undertake a qualitative survey of farmer perceptions towards the importance of the transition period to health and productivity. I’m a strong supporter of consulting end users on their needs/opportunities and of proactively using data to develop hypotheses to test. So, the authors deserve credit for their initiative and endeavors. The authors conclude that, because of their survey results, more research is needed on milk fever/hypocalcemia, in particular, retained placenta, BCS monitoring, peripartum housing facilities, and early lactation calcium supplementation. This is despite the century of research that has already been undertaken in these topic areas. As someone who has worked extensively in this area for a long time, I do not agree with these conclusions and the authors offer no basis for the conclusion other than these diseases/poor metabolic adaptations are regarded as important by Irish dairy farmers. 

AU: Thank you for this comment and acknowledgement of our research endeavours. In our conclusion, which also refers to the reported implementation of management strategies, the required further research is framed within the understanding of the constraints towards the implementation of strategies proven successful in different production systems and those limiting the effectiveness of the management strategies already implemented in the system under study. Dairy production in Ireland offers management opportunities different to those in the context where most of the grazing transition cow research has been conducted (New Zealand). For instance, cows are housed during the dry period and fed a grass-silage based diet. According to data from our resear

---

## [Decision Letter · Decision Letter 1]

19 Nov 2024

Transition cow health and management in pasture-based dairy herds: a farmers’ survey

PONE-D-24-31981R1

Dear Dr. Valldecabres,

We’re pleased to inform you that your manuscript has been judged scientifically suitable for publication and will be formally accepted for publication once it meets all outstanding technical requirements.

Kind regards,

Angel Abuelo, DVM, MRes, MSc, PhD, DABVP (Dairy), DECBHM

Academic Editor

PLOS ONE

Additional Editor Comments (optional):

Reviewers' comments:

Reviewer's Responses to Questions

**Comments to the Author**

1. If the authors have adequately addressed your comments raised in a previous round of review and you feel that this manuscript is now acceptable for publication, you may indicate that here to bypass the “Comments to the Author” section, enter your conflict of interest statement in the “Confidential to Editor” section, and submit your "Accept" recommendation.

Reviewer #1: All comments have been addressed

Reviewer #2: All comments have been addressed

2. Is the manuscript technically sound, and do the data support the conclusions?

Reviewer #1: (No Response)

Reviewer #2: Yes

3. Has the statistical analysis been performed appropriately and rigorously? 

Reviewer #1: (No Response)

Reviewer #2: Yes

4. Have the authors made all data underlying the findings in their manuscript fully available?

Reviewer #1: (No Response)

Reviewer #2: Yes

5. Is the manuscript presented in an intelligible fashion and written in standard English?

Reviewer #1: (No Response)

Reviewer #2: Yes

6. Review Comments to the Author

Reviewer #1: Still some "reporting to"s - eg 339. Another check through may be useful to identify others not picked up

Reviewer #2: (No Response)

7. PLOS authors have the option to publish the peer review history of their article (what does this mean?). If published, this will include your full peer review and any attached files.

Reviewer #1: No

Reviewer #2: No

---

## [Editor Report · Acceptance letter]

25 Nov 2024

PONE-D-24-31981R1 

PLOS ONE

Dear Dr. Valldecabres, 

I'm pleased to inform you that your manuscript has been deemed suitable for publication in PLOS ONE. Congratulations! Your manuscript is now being handed over to our production team.

Kind regards, 

on behalf of

Dr. Angel Abuelo 

Academic Editor

PLOS ONE